# Hydrogel-Graphene Oxide Nanocomposites as Electrochemical Platform to Simultaneously Determine Dopamine in Presence of Ascorbic Acid Using an Unmodified Glassy Carbon Electrode

Jésica Pereyra [1], María V. Martinez [1], Cesar Barbero [1], Mariano Bruno [1] 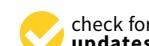 and Diego Acevedo [1,2,*]

1   Instituto de Investigaciones en Tecnologías Energéticas y Materiales Avanzados (IITEMA), Universidad Nacional de Río Cuarto (UNRC)-Consejo Nacional de Investigaciones Científicas y Técnicas (CONICET), Ruta 36 Km 601, Río Cuarto (Córdoba) X5804ZAB, Argentina; jpereyra@exa.unrc.edu.ar (J.P.); mmartinez@exa.unrc.edu.ar (M.V.M.); cbarbero@exa.unrc.edu.ar (C.B.); mbruno@exa.unrc.edu.ar (M.B.)
2   Departamento de Tecnología Química, Facultad de Ingeniería, Universidad Nacional de Río Cuarto, Ruta 36 Km 601, Río Cuarto (Córdoba) X5804ZAB, Argentina
*   Correspondence: dacevedo@ing.unrc.edu.ar; Tel.: +54-358-467-6233

**Abstract:** The detection of dopamine, an important neurotransmitter in the central nervous system, is relevant because low levels of dopamine can cause brain disorders. Here, a novel electrochemical platform made of a hydrogel–graphene oxide nanocomposite was employed to electrochemically determine simultaneously dopamine (DA) and ascorbic acid (AA). Unlike previous work, where the base electrode is modified, the active material (graphene oxide, GO) was dispersed in the hydrogel matrix, making an active nanocomposite where the electrochemical detection occurs. The GO, hydrogel and nanocomposite synthesis is described. Dynamic Light Scattering, UV-visible and FTIR spectroscopies showed that the synthesized GO nanoparticles present 480 nm of diagonal size and a few sheets in height. Moreover, the polymer swelling, the adsorption capacity and the release kinetic of DA and AA were evaluated. The nanocomposite showed lower swelling capacity, higher DA partition coefficient and faster DA release rate than in the hydrogel. The electrochemical measurement proved that both materials can be employed to determine DA and AA. Additionally, the nanocomposite platform allowed the simultaneous determination of both molecules showing two well separated anodic peaks. This result demonstrates the importance of the incorporation of the nanomaterial inside of the hydrogel and proves that the nanocomposite can be used as a platform in an electrochemical device to determinate DA using an unmodified glassy carbon electrode.

**Keywords:** polymer matrix composites; carbon materials; graphene oxide; hydrogel; electrochemical applications

## 1. Introduction

Dopamine (DA) is an important neurotransmitter in the central nervous system [1]. Low levels of DA can cause brain disorders such as schizophrenia and Parkinson's disease [2]. Ascorbic acid (AA), a water soluble antioxidant, is a vital vitamin in the human diet and can protect living organisms from oxidative stress [3]. Both DA and AA coexist in the extracellular fluids of the central nervous system and serum in mammals. However, the AA concentration in these environments is 100–1000 times higher than that of DA. Therefore, the detection of DA in the presence of a large amount of AA has been matter of concern for many researchers [4–7]. Taking into account the aforementioned,

electrochemical methods, such as cyclic voltammetry or linear sweep voltammetry, could be suitable for the simultaneous detection of DA and AA since both molecules are redox active. However, using traditional electrodes, the oxidation peaks of these species overlap, making the simultaneous determination impossible [8–13]. To overcome this problem, various modified electrodes based on graphene have been developed [5,14–16], since GO exhibits good conductivity, wide potential window, large electrochemical active surface area, relatively inert electrochemistry, and high electrocatalytic activity [17,18].

Several nanomaterials has been extensively studied and applied in technological devices such as sensors, due to mainly their novel properties [19–23]. Among them, water-soluble sulfonated graphene [14], chitosan graphene [15], chemically reduced graphene oxide [16,17], and electrochemically reduced graphene oxide (ER-GO) [24] have been developed into modified electrodes. Kim's group developed a modified electrode using polypyrrole and electrochemical reduced graphene oxide (PPy-ER-GO) attaining the selective determination of DA in the presence of AA and uric acid (UA) [25]. Raj et al. fabricated a modified electrode by a self-assembly method employing 1,6-hexanediamine/ER-GO (HDA/ER-GO) [26]. Even though the HDA/ERG-O modified electrode realized the separation of the oxidation peaks of DA, AA, and UA, the toxicity agent HDA may have adverse effects on the performance of the modified electrode. It is noteworthy that, in previous work, the electrode itself is irreversibly modified with the active material.

On the other hand, hydrogels are very versatile soft materials [26–29] and their properties can be easily modified by copolymerization of monomers bearing selected functional groups, which can interact specifically with different compounds of interest [30]. Recently, Martinez et al. [31–33] developed a simple experimental setup where a hydrogel matrix is used to load redox active substances, which they characterized by employing electrochemical techniques. The results show that the redox probes are present as dilute solutions and the hydrogel dimensions fulfill the semi-infinite diffusion boundary conditions, therefore the data analysis can be performed using the theoretical framework for electrochemical measurements in liquid solvents in the semi-infinite regime [31,32]. This is in contrast with modified electrodes where complex data analysis is required [34], since the diffusion is constrained by the finite thickness of the films.

Consider the above mentioned, the incorporation of GO into a soft matrix based on crosslinked acrylamides to obtain a nanocomposite (NanoC) that works as an active material is shown. NanoC offers the following useful advantage: (i) the application to electrochemically determinate DA is straightforward, since it is not necessary to modify the glassy carbon electrode; (ii) the manipulation is easy, because NanoC is a monolithic piece separated from an electrode that can be synthesized in bulk; (iii) NanoC can be independently characterized by: ultraviolet–visible (UV–vis) and FTIR spectroscopies, scanning electron microscopy (SEM), mechanical assays and others [18,35]; and (iv) since the solid NanoC acts as electrolyte, the sampling of analytes from different environments could be separated from the electrochemical detection, allowing the building of portable devices.

In this research work, NanoC was used as a platform to preload and detect simultaneously DA and AA, using a bare glassy carbon without further treatment, by employing electrochemical technique. The nanocomposite interacts selectively with the molecules under study choosing a suitable pH. Partition coefficients higher than 60 (for DA) or 1000 (for AA) at pH 7 (PBS buffer) were achieved. The electrochemical signals of the AA and DA redox molecules, loaded into the gel matrix, overlap [36]. However, two well separated oxidation peaks (at 0.006 V and 0.200 V) were observed using AA and DA preloaded in NanoC. To the best of our knowledge, this is the first time that hydrogel–GO nanocomposites were used to preload and sense electrochemically these molecules. In summary, the NanoC advanced composite can be successfully used to preload DA and AA from complex media, to simultaneously determine DA and AA electrochemically, allowing to build portable electronic devices to make an on-field analysis.

## 2. Materials and Methods

### 2.1. Synthesis

Natural graphite powder (Aldrich) was oxidized using the modified Hummers method [37]. The GO synthesis depicted has been extensively employed, as well as the GO produced has been exhaustively characterized by our research group [18,35,38]. Briefly, graphite powder (2 g) was added to concentrated $H_2SO_4$ (200 mL) immersed in an ice bath. Then, $KMnO_4$ (12 g) was added gradually under constant stirring and cooling to keep the suspension temperature below 20 °C, and the reaction left under stirring for 2 h. Subsequently, 360 mL of distilled water were added ensuring that the temperature does not exceed 50 °C. Then, 40 mL of $H_2O_2$ (a volume fraction of 30%) were added to the mixture, which adopted a bright yellow color. The solid mixture was washed with distilled water until the pH of the suspension reached a value of 3. The suspension was centrifuged (30 min at 7000 rpm) and the supernatant discarded. Then, the graphite oxide was dispersed in distilled $H_2O$ and subjected to ultrasound for 1 h to generate GO sheets or monolayers [38].

### 2.2. Graphene Oxide Characterization

Fourier transform infrared (FTIR) spectra of the GO were recorded from KBr pellets of solid samples in a Nicolet Impact 410 spectrometer. The UV–Vis spectra of GO water dispersion were recorded on a HP 8452A spectrophotometer. The Dynamic Light Scattering (DLS) measurements were performed in a Malvern 4700 DLS with goniometer and a 7132 correlator, with light of an argon-ion laser operating at 488 nm. The measurement of a water dispersion of GO (0.5 mg/mL) was made at the scattering angle of 90°.

### 2.3. Nanocomposites and Hydrogels Synthesis

The hydrogel platform (PAAm) was synthesized by free radical polymerization in cylindrical molds. Acrylamide (AAm, 0.071 g/mL) was employed as monomer and N,N-methylene bisacrylamide (BIS, 0.310 mg/L) as a crosslinking agent. The polymerization was initiated by a redox initiator system: ammonium persulfate (APS, 1 μg/mL) and tetramethylene diamine (TEMED, 1 μL/mL). AAm, BIS, and APS were first dissolved in 4 mL of PBS buffer (pH 5.8) to constitute a pre-gelling mixture (PGM). After that, TEMED was added to the PGM to produce the polymerization and crosslinking in 24 h. The nanocomposites (NanoC) of polyacrylamide hydrogels loaded with GO (PAAm-GO) were synthesized using the same procedure but adding 20 mg of dispersed GO to the 4 mL of PGM to produce PAAm-GO, prior to the TEMED addition [39]. The nanocomposites PAAm-GO obtained using this procedure has been previously characterized and reported by our research group [18]. Then, PAAm and the NanoC cylinders were cut into slices, washed several times with distilled water and dried for 48 h at room temperature under moderate vacuum (10,130 Pa). The samples were stored in a desiccator for later use.

### 2.4. Characterization of Hydrogels and Nanocomposites

#### 2.4.1. Fourier Transform Infrared (FTIR)

Fourier transform infrared (FTIR) spectra of the PAAm and PAAm-GO were recorded from KBr pellets of solid samples in a Bruker Tensor 27 spectrometer.

#### 2.4.2. Swelling

The swelling kinetics of PPAm and NanoC were evaluated by measuring water absorption into the materials [4]. Dried hydrogels, previously weighed, were placed in pure water and kept at constant ambient temperature. The samples were removed at regular time intervals. The surface water was wiped off with filter paper. Then, the samples were weighed and placed back into the water to

continue the swelling process. The measurements were performed until a constant weight was reached. The swelling percentage (% Sw) was calculated according to Equation (1), as a function of the time.

$$\% \, Sw = \frac{w(t) - w_o}{w_o} \times 100 \tag{1}$$

where w(t) represents the weight of hydrogel in swollen state at time t and $w_o$ is the weight of the dry hydrogel. All the measurements were performed in triplicate.

The initial swelling rate was determined using the slope of the first points of the kinetic graph. The % Sw at equilibrium (% $Sw_{eq}$) was determined using the value of the weight achieved at two days (2880 min) when the weight of the hydrogels achieved a constant value.

### 2.4.3. Absorption Capacity of Solutes

The equilibrium distribution of the solute between a solid material (gel or NanoC) and the aqueous phase can be estimated using the partition coefficient (Pc), which is calculated using Equation (2).

$$Pc = \frac{C_{in}}{C_{aq}} \tag{2}$$

where $C_{in}$ and $C_{aq}$ are the molar concentration of the chemical substance under study in the gels or NanoC and in the aqueous solution, respectively [5]. In this work, dopamine (Silgma-Aldrich) and ascorbic acid (Cicarelli) absorption capacity in NanoC and PAAm was evaluated at pH 3, 7 and 9. Before the Pc determinations, DA and AA solutions of different concentrations were made, dissolving the adequate quantity of the solute in buffer solutions (pH 3, 7 and 9). Then, the absorbance of DA (at $\lambda$ = 280 nm) and AA (at $\lambda$ = 265 nm) solutions versus the concentration of these solutions were measured and the extinction molar coefficients of DA and AA at pH 3, 7 and 9 were calculated.

Experimentally, a piece of known dry mass of NanoC or PAAm was immersed in 3 mL of the dopamine (DA) or ascorbic acid (AA) solutions at a known concentration. After 48 h immersion, the concentration of the remaining solution was determined by UV-visible spectroscopy (Hewlett-Packard-8453 UV-visible spectrophotometer, Agilent Technologies Deutschland GmbH, Boblingen, Germany), and the volume of the hydrogel was determined using gravimetric measurements. Finally, the Pc was determined using Equation (2).

### 2.4.4. Kinetic of Solute Release

The DA and AA released from the NanoC and PAAm were analyzed by monitoring the released drug absorbance as function of the time. Experimentally, a cylindrical weighed piece of NanoC or PAAm, previously loaded with DA or AA (48 h) was immersed in 10 mL of distilled water (at 20 °C), the concentration of the studied molecules in the solution at regular times C(t) (during 24 h) was determined by UV-Visible spectroscopy and at the same time intervals the mass of the matrix was measured. All measurements were performed in triplicate. Using these data, the initial release rate ($\upsilon$) was estimated as the derivate of the concentration vs. time in the initial stages of the release process.

### 2.4.5. Electrochemical Measurements

All electrochemical experiments were performed using a computer-controlled potentiostat (Autolab PGSTAT30, Ecochemie, Metrohm AG, Herisau, Switzerland). The electrochemical studies were performed by experiments cyclic voltammetry (CV) and linear sweep voltammetry (LSV), in a three-electrode cell, as shown in Scheme 1. A glassy carbon electrode (GC), a Pt grid and a Ag/AgCl (KCl sat. solution) were used as working electrode, counter electrode and reference electrode (RE), respectively. The electrochemical behavior of DA and AA was studied using different concentrations in phosphate buffer (pH 7). Dopamine (0.25 mM), Dopamine (1 mM), Ascorbic Acid (0.25 mM), Ascorbic Acid (1 mM), Dopamine (0.25 mM) and ascorbic acid (0.25 mM), and Dopamine (1 mM) and

Ascorbic Acid (1 mM) were employed in the electrochemical measurements. Before the electrochemical measurements, the matrices (PAAm and NanoC, d = 10 mm; $d_p$ = 3 mm) were pre-loaded with DA and AA during 24 h at 20 °C. Then, the matrices were poured on a Pt grid inside of the electrochemical cell (see Scheme 1). Then, the unmodified glassy carbon working electrode was gently pressed onto the hydrogel and the cell was filled with the buffer solution (PH 7). The potential window employed for CV and LSV techniques, was from −0.2 to 0.6 V. The LSVs were performed at a scanning speed of 1 mV/s, and the concentration of the study solutions was of 1 mM in 0.1 M phosphate buffer at pH 7.

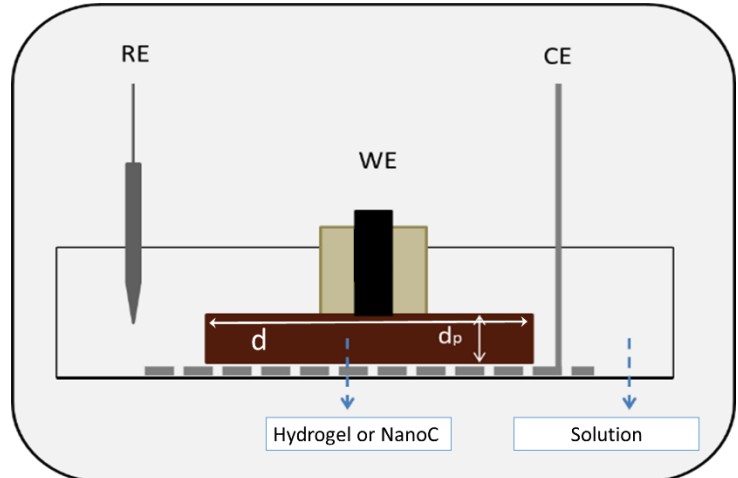

**Scheme 1.** Schematic representation of the set-up used to carry out the electrochemical measured using preloaded hydrogels and NanoC. RE, reference electrode; WE, working electrode; CE, counter electrode; d, the matrix diameter; $d_p$, the matrix height.

## 3. Results

### 3.1. Synthesis and Characterization

This polymeric gels based on polyacrylamide used for sorption of DA and AA organic molecules were characterized by infrared spectroscopy.

The FTIR spectra of the PAAm and NanoC are shown in Figure 1a.

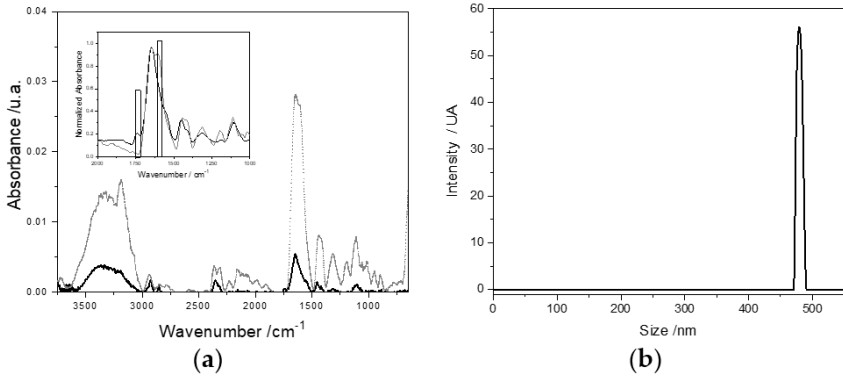

**Figure 1.** (**a**) FTIR of the hydrogel PAAm (black) and the NanoC (PAAm-GO, gray); and (**b**) Dynamic Light Scattering (DLS) of a graphene oxide (GO).

The GO synthesis depicted in the NanoC preparation has been extensively employed as well as exhaustively characterized in previous work of our research group [18,35,38]. Following the synthetic procedure reported, it is possible to obtain GO, a very oxidized material [40], with a ID/IG ratio (measured by Raman spectroscopy) of ca. 0.8 [38]. Moreover, it has been shown that an ID/IG ratio of

0.8 corresponds to a C/O ratio of 1.8 obtained from the curve fitting of XPS spectra [41]. Therefore, the GO obtained presents a high degree of oxygenated moieties. In addition, the size of the GO measured by HRTEM presented small laminates very well distributed [38], with a mean DLS apparent size of ca. 480 nm (Figure 1b).

The kinetic swelling kinetic is an important parameter to evaluate not only the capacity of the hydrogel and the NanoC to absorb water but also the amount of crosslinking in the 3D network. Figure 2 shows the hydrogel and the composite material swelling kinetics.

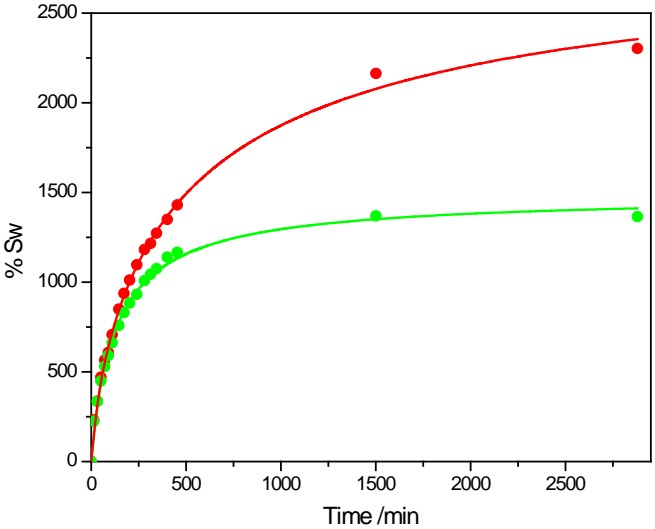

**Figure 2.** Swelling kinetic of (●) PAAM and (●) NanoC.

It is apparent that both the maximum value of %Sw and initial swelling rate of the hydrogel matrix (PAAm, %Sw = 2300 ±180, swelling rate 16.52 ± 0.11 %Sw/min) are larger than the ones measured in NanoC (%Sw = 1540 ± 130, swelling rate 16.01 ± 0.12 %Sw/min). It is likely that GO nanoparticles increase the crosslinking of the matrix, thus reducing swelling.

### 3.2. NanoC and Hydrogel Absorption Capacity.

The PAAm hydrogel and NanoC capacities to absorb DA and AA molecules was studied by UV-visible spectroscopy following the intensity of the characteristic absorption bands of the molecules at three different pH values (3, 7, and 9). The extinction coefficients ($\varepsilon$) calculated and at the maxima wavelength employed are depicted in Table 1.

**Table 1.** Extinction coefficients at pH 3, 7 and 9 at the maximum wavelength of DA and AA.

| Maximum Wavelength | DA | | | AA | | |
|---|---|---|---|---|---|---|
| Extinction Coefficients | pH 3 | pH 7 | pH 9 | pH 3 | pH 7 | pH 9 |
| $\varepsilon \times 10^{-3}$ (L/(cm·mol)) | 2.31 | 3.02 | 3.99 | 9.710 | 17.75 | 5.68 |
| $\lambda$ (nm) | 280 | 280 | 293 | 244 | 265 | 266 |

Using the calculated extinction coefficients, the molar concentration of each studied molecule inside and outside of the polymers was calculated and the partition coefficient of the gels determined using Equation (2). The results are reported in Table 2.

**Table 2.** Physicochemical parameters: Partition coefficients of DA and AA in PAAm and NanoC materials, Equilibrium Swelling Percentages in DA and AA solution (%Sw$_{eq}$) at pH 7 and initial release rates ($\upsilon$).

| Material | Partition Coefficient (Pc) | | | | | | % Sw$_{eq}$ | | $\upsilon$ (10$^9$) M/(min·cm$^2$) | $\upsilon$ (10$^9$) M/(min·cm$^2$) |
| | DA | | | AA | | | | | | |
| | pH 3 | pH 7 | pH 9 | pH 3 | pH 7 | pH 9 | DA | AA | DA | AA |
|---|---|---|---|---|---|---|---|---|---|---|
| PAAm | 12.38 | 25.67 | 16.69 | 84.15 | 1084.10 | 75.09 | 1334 ± 79 | 1353 ± 143 | 226.23 | 16.10 |
| NanoC | 7.93 | 65.26 | 6.83 | 186.40 | 946.22 | −5.90 | 1436 ± 177 | 1195 ± 84 | 156.9 | 6.49 |

## *3.3. Release Kinetic of DA and AA*

The release kinetics of DA and AA molecules were studied at pH 7 since at this pH value the materials present the best adsorption capacity (Figure 3).

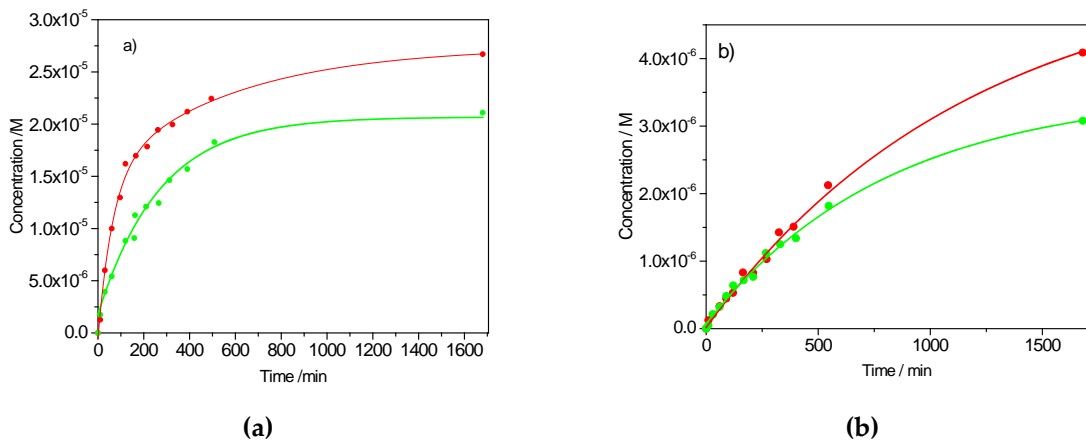

**Figure 3.** Release kinetics of (**a**) DA and (**b**) AA from the PAAm (●) and NanoC (●).

Using the experimental kinetics data (concentration vs. time), it was possible to calculate the initial release rate ($\upsilon$) as the derivate of the concentration vs. time in the initial stages of the release process (Table 2).

## *3.4. Electrochemistry Measurements*

To evaluate the electrochemical behavior of both composites, unloaded analytes were studied. In Figure 4, the electrochemical response of these matrices using a potential window between 0 and 0.8 V in 1 M H$_2$SO$_4$ is shown.

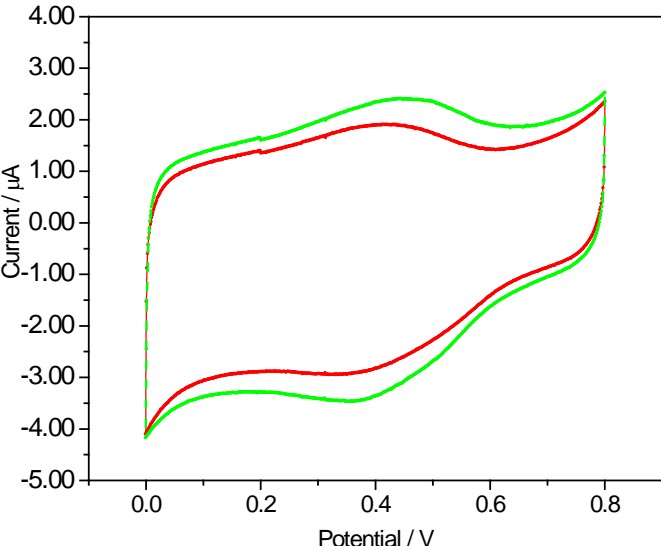

**Figure 4.** Cyclic Voltammogram of PAAm (red), and NanoC (green) in 1 M $H_2SO_4$, scan rate 50 mV/s, glassy carbon working electrode, Ag/AgCl reference electrode and platinum as counter-electrode.

The electrochemical responses (CV at 1 mV/s) of DA and AA preloaded in on different pieces of PAAm and NanoC (from solutions of 0.25 mM at pH 7 (PBS)) were evaluated (Figure 5).

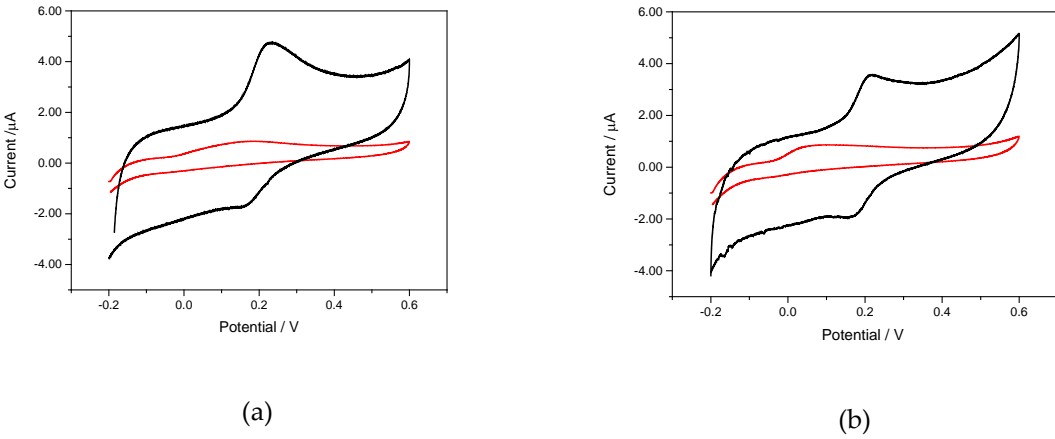

(a)　　　　　　　　　　　　　　　　　　　　　　　(b)

**Figure 5.** Voltammetric response of preloaded (**a**) PAAm and (**b**) NanoC with DA (back line) and AA (red line) from respective solution 0.25 mM in PBS pH 7. Scan rate: 10 mV/s.

The electrochemical response of PAAm and NanoC LSV was carried out using the preloaded matrices with a solution of 1 mM DA and 1 mM AA in PBS (pH 7), at scan rate of 1 mV/s (Figure 6).

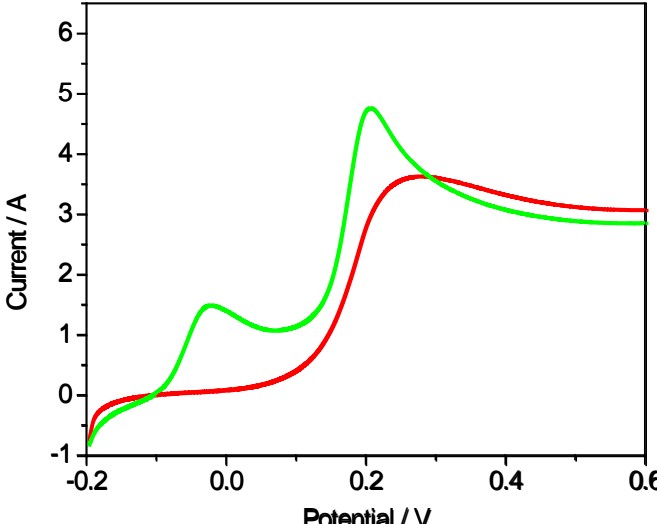

**Figure 6.** Linear sweep voltammetry of preload PAAm (red) and NanoC (green) from a mixture containing AA (1 mM) and DA (1 mM) in PBS (pH 7). Scan rate: 1 mV/s.

## 4. Discussion

The FTIR spectrum of PAAm presents the characteristic bands of the polymer. The bands at 2881 and 2974 cm$^{-1}$ were assigned to the –CH$_3$ symmetric and asymmetric stretching, and the band at ca. 3500 cm$^{-1}$ corresponds to N-H stretching of the secondary amides (Figure 1a). Moreover, the band at 1650 cm$^{-1}$ is characteristic of the C=O stretching vibration of the amide I band and the bandat 1570 cm$^{-1}$ corresponds to the amide II band [42]. On the other hand, the NanoC presents not only the typical bands of the polymer but also bands at 1750 cm$^{-1}$ and 1630 cm$^{-1}$ that can be assigned to the functional group (>C=O) present in GO [43] assuring the presence of GO inside the hydrogel (Figure 1a).

The DLS size distribution in Figure 1b shows that the GO particles have a mean size of 480 nm. This result is in agreement with the data published by Pereyra et. al., who used the same synthetic procedure [38]. The results obtained from characterization indicate GO synthesized present the same physical and chemical properties that previously published [18,35,38].

Figure 2 shows the hydrogel and composite material swelling kinetics in pure water. The equilibrium swelling of PAAm is ca. 2300 ± 180% while for NanoC the swelling percentage is ca. 1584 ± 130%. The nanocomposite equilibrium swelling value is lower than that of the hydrogel. It seems that the NanoC elastic properties are lowered due to the incorporation of the nanomaterials inside of the three-dimensional matrix of the hydrogel. The diminution of the swelling is attributed mainly to the hydrogen bonding that can be formed between the GO hydroxyl and carboxyl groups with the oxygenated functional group of the PAAm. These new interactions produce an enhancement of the chemical crosslinking, adding a physical crosslinking resulting in the swelling equilibrium reduction [35,42,44,45]. The swelling in the equilibrium of the nanocomposite NanoC decreases compared to PAAm. However, the initial swelling rate of 16.01 ± 0.12 and 16.52 ± 0.11 % Sw/min, respectively, suggests a similar hydrophilicity of these materials. Although the swelling behavior of PAAm and NanoC in water is markedly different (Figure 2), the swelling in AA and DA solutions, at pH 7, presents similar values in both matrices (see Table 2).

Table 2 shows that at pH 3 and 9 the Pc values decrease for both molecules in the PAAm and NanoC materials compared to the studies conducted at pH 7. This behavior can be attributed to the electrostatic repulsion between the charges in the hydrogel and in the studied molecules at this pH value. Therefore, the variation of the Pc with the pH is relevant because it allows adjusting the sorption capacity of the molecules in an easy manner. Moreover, the study shows that AA presents higher Pc than DA regardless of the material tested except at pH 9, probably due to the presence of carboxylate

ion generated at this pH on the GO nanosheets. In addition, NanoC presents lower sorption capacity than PAAm hydrogel. This behavior is in agreement with the lower swelling percentage of this material compared with the pristine hydrogel. Considering the higher values AA of Pc obtained at pH 7, it is possible to conclude that the AA presents more affinity to both matrices than DA, probably due to the functional groups interacting better with the chemical structure of the matrix. This result is in total agreement with the lower initial rate values of AA in comparison with DA (AA initial rate value is ca. 10-fold lower than the υ of DA inside of NanoC supporting the previous result, Table 2) [46,47].

The concentration of the studied molecules (DA and AA) released in function of the time. Comparing the maximum concentration released by each matrix, it is possible to conclude that NanoC matrix retains not only DA but also AA more efficiently than PAAm, as shown in Figure 3. In addition, in Figure 3b, it can be observed that AA is retained during more time inside of the matrices than the DA, which is in total agreement with the PC values previously determined.

As can be observed in Table 2, the initial release rate values allow confirming that inside of the NanoC matrix not only DA but also AA presents a lower initial rate than in PAAm matrix, which confirms that the GO functional groups allow a better chemical interaction of the small molecules with the 3D structure.

Besides, the DA initial release rate inside both matrices presents similar values; the AA initial release rate value in NanoC matrix is more than 2.5-fold lower than in PAAm matrix, indicating that NanoC presents more interaction with AA molecules, allowing to ensure that NanoC could be used as a selective matrix, allowing the easy diffusion of DA and retaining AA.

The current response profile in Figure 4 reveals a rectangular shape with a broad band around 0.4 V, which is the typical response of carbon electrode with surface electroactive redox groups (quinone-like moieties). The increase in the current observed in the CV of GC in contact with NanoC can be attributed to the increased surface area due to the contribution of GO electrically connected with the pressing glassy carbon electrode.

Considering that DA is a relevant molecule that regulates functions of the central nervous system, its detection is of great interest. However, it is known that not only DA but also AA is present in the extracellular fluid of the central nervous system, and in higher concentration [48]. Moreover, the electrochemical determination of DA becomes difficult because the AA oxidation potential is close to DA oxidation potential when solid unmodified electrodes are used [49]. Kim et. al showed that using vitreous carbon electrodes modified with a deposit of GO can separate the anodic peaks corresponding to DA and AA oxidation. In this manner, the electrochemical determination of each compound is possible [16].

The electrochemical profile of DA (Figure 5) inside of both platforms presents a typical quasi-reversible oxidation-reduction similar to the DA in solution at ca. 0.2 V [48]. In addition, the electrochemical response of AA (inside of both platforms) shows the anodic peak, even though the cathodic peak is not present. This results indicate that the AA oxidation reaction is coupled with an irreversible chemical reaction [24,48]. Moreover, the oxidation peak of AA preloaded in PAAm (ca. 0.2 V) shifts to a more negative oxidation value (0.06 V) than when it is loaded inside of NanoC. Thus, the difference between DA and AA oxidation peaks reach a value of 0.14 V, which shows the importance of the NanoC allowing discriminate both molecules [50].

In Figure 6, the linear sweep voltammetry of each preloaded platform is presented; only one peak can be observed when the analytes (DA and AA) are preloaded in PAAm platform. Nevertheless, two well-defined and completely resolved anodic peaks at ca. 0.006 V and 0.2 V can be observed when both molecules are preloaded in the NanoC platform, corresponding to the oxidation of AA and DA, respectively. The good difference of the peak-to-peak potential for DA and AA could be attributed to the different adsorption affinity of these compounds inside of the NanoC structure [51]. In NanoC the AA partition coefficient is higher than DA, implying a greater concentration of the AA redox species within the nanocomposite. However, it is not possible to appreciate differences in the anodic peak potential; this behavior can be explained by taking into account the faster DA

diffusion inside of the platform (see Table 2 initial release rate value). These results demonstrate that the GO presence in the composite produce a separation of AA and DA oxidation peaks. In addition, it confirms that the electrochemical detection of both molecules present simultaneously is possible using the NanoC platform. Moreover, the difference between the oxidation potentials of both species is 0.194 V, suggesting that a selective measurement of one component in the presence of the other or the simultaneous determination of both analytes preloaded in NanoC is feasible.

## 5. Conclusions

This study shows that is possible to determinate dopamine in presence of ascorbic acid by using an unmodified GC electrode in a preloading PAAm-GO nanocomposite. Moreover, an easy production of the nanocomposite has been shown, starting with the synthesis of GO, followed by radical polymerization of acrylamide in presence of the GO dispersion.

The characterization of the GO particles showed a mean size average of 480 nm, made of few 2D grapheme sheets. In addition, the swelling of the hydrogel decreased by the incorporation of GO, from 2300% to 1584%, probably due to the interaction of the oxidized groups in GO with the functional groups (-NH$_2$) present in the polymer. The partition coefficients of DA and AA inside the matrices showed that the AA presents a better interaction than DA. The release kinetics studies of DA and AA molecules showed that NanoC matrix retains not only DA but also AA more efficiently than PAAm. The electrochemical behavior of these matrices evaluated by CV revealed a rectangular shape with a broad band around 0.4 V, which a typical response of a carbon electrode with surface electroactive groups. In addition, the electrochemical response of DA and AA preloaded separately using the PAAm and NanoC platforms showed that using the nanocomposite the AA peak oxidation potential shifted to a more negative value, allowing the simultaneous determination of DA in presence of AA. By using LSV, it was possible to observe a good oxidation peak separation (0.194 V), suggesting that a selective measurement of DA and AA can be carried out. All these results support that it is possible to determine DA in presence of AA employing an unmodified GC electrode, preloading the molecules inside of NanoC platform.

**Author Contributions:** D.A. and M.B. planned and supervised the work. J.P. and M.V.M. performed the experimental work and data analysis. D.A. and C.B. analyzed the data and wrote the manuscript.

**Funding:** This research was funded by CONICET PIP 2014-2016, SECYT-UNRC PPI 2016-2018, and FONCYT PICT 2013-2716 and 2016/1706.

**Acknowledgments:** Diego Acevedo, Mariano Bruno, and Cesar Barbero are permanent research staff of CONICET. María V. Martinez and Jésica Pereyra thank CONICET for a graduate fellowship.

**Conflicts of Interest:** The authors declare no conflict of interest.

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
