# Peer review of "Hydrogel-Graphene Oxide Nanocomposites as Electrochemical Platform to Simultaneously Determine Dopamine in Presence of Ascorbic Acid Using an Unmodified Glassy Carbon Electrode"

_jcs, doi:10.3390/jcs3010001_

Reviewer 1 Report

this is an interesting work, the Hydrogel-Graphene Oxide Nanocomposites were well synthesized. we think it can be pubilshed after major revision.

XPS is necessary to provided.

bet is also needed. 

Some relevant paper  should be cited : Applied Catalysis B: Environmental 240 (2019) 92–101; Journal of Catalysis 361 (2018) 238–247; Journal of Catalysis 355 (2017) 1–10; Carbon 122 (2017) 287-297; J. Mater. Chem. A, 2018,6, 20304-20312. 

Author Response

We deeply appreciate the reviewers’ recommendations to our manuscript and for giving us the opportunity to improve it.

1) XPS is necessary to provided.

We appreciate the recommendation, the XPS spectroscopy is a powerful technique that could improve our research work in order to determinate the oxidation degree of the GO synthesized. However, this is out of our scope in this work. The synthetic method employed to produce GO has been used by our research group several times, and the characterization of the GO synthetized in this manner always presented the same physicochemical properties that has been exhaustively studied and already published in [1–3]. Nevertheless, in order to improve the manuscript, taking into account the reviewer suggestion, the following text has been incorporated in the manuscript (Section 3.1): 

“The GO synthesis depicted in the NanoC preparation has been extensively employed as well as exhaustively characterized in our research group [1–3]. Follow the synthetic procedure reported by raman spectroscopy is possible to obtain GO with a ID/IG ratio ca. 0.8 [2], that resemble a very oxidized material [4]. Moreover, it has been shown that an ID/IG ratio of 0.8 corresponds to a C/O ratio of 1.8 obtained from the curve fitting of XPS spectra [5]. Therefore, the GO obtained presents a high degree of oxygenated moieties. Also, the size of the GO measured by HRTEM presented small laminates very well distributed [2], with a 480 nm of diagonal size (Figure 1b)” 

2) Bet is also needed.

We appreciate the reviewer’s comment. However, in order to perform nitrogen adsorption isotherm, the material should be dried. The area and the porosity of the NanoC obtained depend on the drying method used. Also, this specific area will not be representative of the wet hydrogel (that is the matrix used in the dopamine/ascorbic acid determination). Clearly, the properties measured by area BET will not represent the specific area or the porosity of the nanocomposite when it is used to preload and sense electrochemically dopamine and ascorbic acid.  

3) Some relevant paper should be cited: Applied Catalysis B: Environmental 240 (2019) 92–101; Journal of Catalysis 361 (2018) 238–247; Carbon 122 (2017) 287-297; - J. Mater. Chem. A, 2018,6, 20304-20312. 5- Journal of Catalysis 355 (2017) 1–10;

In order to improve the manuscript, the relevant publications that the reviewer suggested, have been incorporate to the main text.

References

[1]       L. Mulko, J.Y. Pereyra, C.R. Rivarola, C.A. Barbero, D.F. Acevedo, Improving the retention and reusability of Alpha-amylase by immobilization in nanoporous polyacrylamide-graphene oxide nanocomposites, Int. J. Biol. Macromol. (2018). doi:https://doi.org/10.1016/j.ijbiomac.2018.09.078.

[2]       J.Y. Pereyra, E.A. Cuello, H.J. Salavagione, C.A. Barbero, D.F. Acevedo, E.I. Yslas, Photothermally enhanced bactericidal activity by the combined effect of NIR laser and unmodified graphene oxide against Pseudomonas aeruginosa, Photodiagnosis Photodyn. Ther. 24 (2018) 36–43. doi:https://doi.org/10.1016/j.pdpdt.2018.08.018.

[3]       J.Y. Pereyra, E.A. Cuello, R.C. Rodriguez, C.A. Barbero, E.I. Yslas, H.J. Salavagione, D.F. Acevedo, Synthesis and characterization of GO-hydrogels composites, IOP Conf. Ser. Mater. Sci. Eng. 258 (2017) 12002. http://stacks.iop.org/1757-899X/258/i=1/a=012002.

[4]       G.K. Ramesha, S. Sampath, Electrochemical Reduction of Oriented Graphene Oxide Films: An in Situ Raman Spectroelectrochemical Study, J. Phys. Chem. C. 113 (2009) 7985–7989. doi:10.1021/jp811377n.

[5]       A. Pulido, P. Concepción, M. Boronat, C. Botas, P. Alvarez, R. Menendez, A. Corma, Reconstruction of the carbon sp2 network in graphene oxide by low-temperature reaction with CO, J. Mater. Chem. 22 (2012) 51–56. doi:10.1039/c1jm14514b.

Reviewer 2 Report

It is good enough to publish in this journal. However, correction of spelling is needed.

For example, line 19  Uv-visible ... --> UV-visible ...

                      line 177 This Polymeric gels ...  --> This polymeric ....

Author Response

We deeply appreciate the Reviewer’s positive evaluation of our manuscript.

We have reviewed the text with a native speaker correcting spelling and grammar mistakes in line 19 and line 177 among others, meticulously,

Reviewer 3 Report

This review addresses the manuscript presented by Acevedo and coworkers that is entitled “Hydrogel-Graphene Oxide Nanocomposites as an Electrochemical Platform…” that is submitted for publication to Journal of Composites Science.

The article presents new work on the use of a GO-modified hydrogel for voltammetric analysis of dopamine in the presence of ascorbic acid. There is a substantial amount of literature on voltammetric analysis of dopamine in the presence of common, electroactive molecules such as AA. The literature covered by the authors is quite minimal in light of this. A more thorough listing of references in this area should be completed by the authors prior to publication.

The article also does not read well. While the language issues are not debilitating they are sufficiently serious as to require substantial rewriting prior to publication. These issues begin in the title and run throughout the article. The unique aspect appears to be the use of the GO-modification of the hydrogel to produce a system that can resolve the DA and AA voltammetric peaks without requiring modification of the working electrode. But the flow of the article is cumbersome. Also, the article indicates that the thickness of the DO nanoparticles is measured but I could not find the data and experiments to support this in the manuscript. I assume that this measured by AFM?

These issues need to be addressed prior to publication

Author Response

REPONSE REVIEWER #3:

We appreciate your prompt response with the reviewers’ recommendations to our manuscript and for giving us the opportunity to improve it.  

1-    There is a substantial amount of literature on voltammetric analysis of dopamine in the presence of common, electroactive molecules such as AA. The literature covered by the authors is quite minimal in light of this. A more thorough listing of references in this area should be completed by the authors prior to publication.

We introduce several publications related with the electrochemical detection of DA and AA. [6–11]

2-    The article also does not read well. While the language issues are not debilitating they are sufficiently serious as to require substantial rewriting prior to publication. These issues begin in the title and run throughout the article.

We have reviewed the text with a native speaker correcting spelling and grammar mistakes meticulously. Also, the title has been modified in order to attend the recommendation.

3-    The unique aspect appears to be the use of the GO-modification of the hydrogel to produce a system that can resolve the DA and AA voltammetric peaks without requiring modification of the working electrode. But the flow of the article is cumbersome.

Taking into account the suggestion we make several changes in the manuscript also we incorporated more bibliography to clarified the flow of the article.

4-    Also, the article indicates that the thickness of the DO nanoparticles is measured but I could not find the data and experiments to support this in the manuscript. I assume that this measured by AFM?

The reviewer is right, however, we would like to emphasize that the aim of the manuscript is to show the electrochemical potential of the polymeric gel Nanocomposite to sense two important molecules. The synthetic method employed to produce GO has been used by our research group several times, and the characterization of the GO synthetized in this manner always presented the same physicochemical properties that has been exhaustively studied and already published in [1–3].  In order to improve the manuscript, taking into account the reviewer suggestion, the following text has been incorporated in the manuscript (Section 3.1): “The GO synthesis depicted in the NanoC preparation has been extensively employed as well as exhaustively characterized in our research group [1–3]. Follow the synthetic procedure reported by raman spectroscopy is possible to obtain GO with a ID/IG ratio ca. 0.8 [2], that resemble a very oxidized material [4]. Moreover, it has been shown that an ID/IG ratio of 0.8 corresponds to a C/O ratio of 1.8 obtained from the curve fitting of XPS spectra [5]. Therefore, the GO obtained presents a high degree of oxygenated moieties. Also, the size of the GO measured by HRTEM presented small laminates very well distributed [2], with a 480 nm of diagonal size (Figure 1b)”

 References 

[1]       L. Mulko, J.Y. Pereyra, C.R. Rivarola, C.A. Barbero, D.F. Acevedo, Improving the retention and reusability of Alpha-amylase by immobilization in nanoporous polyacrylamide-graphene oxide nanocomposites, Int. J. Biol. Macromol. (2018). doi:https://doi.org/10.1016/j.ijbiomac.2018.09.078.

[2]       J.Y. Pereyra, E.A. Cuello, H.J. Salavagione, C.A. Barbero, D.F. Acevedo, E.I. Yslas, Photothermally enhanced bactericidal activity by the combined effect of NIR laser and unmodified graphene oxide against Pseudomonas aeruginosa, Photodiagnosis Photodyn. Ther. 24 (2018) 36–43. doi:https://doi.org/10.1016/j.pdpdt.2018.08.018.

[3]       J.Y. Pereyra, E.A. Cuello, R.C. Rodriguez, C.A. Barbero, E.I. Yslas, H.J. Salavagione, D.F. Acevedo, Synthesis and characterization of GO-hydrogels composites, IOP Conf. Ser. Mater. Sci. Eng. 258 (2017) 12002. http://stacks.iop.org/1757-899X/258/i=1/a=012002.

[4]       G.K. Ramesha, S. Sampath, Electrochemical Reduction of Oriented Graphene Oxide Films: An in Situ Raman Spectroelectrochemical Study, J. Phys. Chem. C. 113 (2009) 7985–7989. doi:10.1021/jp811377n.

[5]       A. Pulido, P. Concepción, M. Boronat, C. Botas, P. Alvarez, R. Menendez, A. Corma, Reconstruction of the carbon sp2 network in graphene oxide by low-temperature reaction with CO, J. Mater. Chem. 22 (2012) 51–56. doi:10.1039/c1jm14514b.

[6]       H. Cong, B. Yu, X. Zhang, S. Yang, Electrochemical detection of dopamine using carbon nanomaterial based electrodes, 2016.

[7]       R. Walsh, U. Ho, X.L. Wang, M.C. DeRosa, Selective dopamine detection using aptamer-functionalized glassy carbon electrodes, Can. J. Chem. 93 (2015) 572–577. doi:DOI 10.1139/cjc-2014-0444.

[8]       N. Tukimin, J. Abdullah, Y. Sulaiman, Review—Electrochemical Detection of Uric Acid, Dopamine and Ascorbic Acid, J. Electrochem. Soc. 165 (2018) B258–B267. doi:10.1149/2.0201807jes.

[9]       Z. Chang, Y. Zhou, L. Hao, Y. Hao, X. Zhu, M. Xu, Simultaneous determination of dopamine and ascorbic acid using β-cyclodextrin/Au nanoparticles/graphene-modified electrodes, Anal. Methods. 9 (2017) 664–671. doi:10.1039/C6AY03013K.

[10]     E.C. Ilinoiu, F. Manea, P.A. Serra, R. Pode, Simultaneous/selective detection of dopamine and ascorbic acid at synthetic zeolite-modified/graphite-epoxy composite macro/quasi-microelectrodes, Sensors (Switzerland). 13 (2013) 7296–7307. doi:10.3390/s130607296.

[11]     S. Qi, B. Zhao, H. Tang, X. Jiang, Determination of ascorbic acid, dopamine, and uric acid by a novel electrochemical sensor based on pristine graphene, Electrochim. Acta. 161 (2015) 395–402. doi:10.1016/j.electacta.2015.02.116.

Round  2

Reviewer 1 Report

It is well revised and can be accepted now.

J. Compos. Sci. EISSN 2504-477X Published by MDPI AG, Basel, Switzerland RSS E-Mail Table of Contents Alert
Back to Top